# A Novel FMCW Radar Scheme with Millimeter Motion Detection Capabilities Suitable for Cardio-Respiratory Monitoring

**DOI:** 10.3390/s25092765

**Published:** 2025-04-27

**Authors:** Orlandino Testa, Renato Cicchetti, Stefano Pisa, Erika Pittella, Emanuele Piuzzi

**Affiliations:** Department of Information Engineering, Electronics and Telecommunications, Sapienza University of Rome, Via Eudossiana 18, 00184 Rome, Italy; orlandino.testa@uniroma1.it (O.T.); renato.cicchetti@uniroma1.it (R.C.); stefano.pisa@uniroma1.it (S.P.); erika.pittella@uniroma1.it (E.P.)

**Keywords:** FMCW radars, vital sign monitoring, remote monitoring

## Abstract

A new modulation scheme for frequency-modulated continuous-wave (FMCW) radars with millimeter-level target motion detection capability is presented. The proposed radar scheme is free from the synchronization constraint and exhibits low sensitivity to internal parasitic mutual coupling, thus significantly reducing its design complexity without worsening its performance in terms of accuracy and operating ranges. Alternatively to canonical FMCW radars, which exploit chirp signals with triangular or sawtooth-like frequency variation, a radar based on a sinusoidal frequency modulation, which does not require specific synchronization procedures to achieve accurate motion detection even at a short distance from the radar, was developed. Both numerical and experimental results, performed with a 24 GHz radar, have shown the suitability of the proposed modulation scheme for monitoring very small target movements, consistent with those typically exhibited by the human thorax during basic vital activities (heartbeat and respiration). This makes the proposed radar scheme a suitable solution for contactless heart and breath rate monitoring.

## 1. Introduction

Short-range radars are widely used in the automotive field, through-the-wall target detection, environmental localization, and biomedical applications [1,2,3]. Particularly relevant is in the biomedical area, where radar systems have been proposed for the localization of children and elderly people and for the remote monitoring of vital signs such as respiratory and cardiac activity [4,5,6,7,8,9]. For these and other applications, ultrawideband (UWB), Doppler continuous wave (DCW), and frequency-modulated continuous wave (FMCW) radars have been used, with the latter being the main radar sensor solutions [6].

DCW radars are characterized by simple hardware architectures based on homodyne or heterodyne demodulation techniques [6]. Both DCW solutions can achieve high accuracy in target motion detection, but they are not suitable for range discrimination to reduce environmental cluttering and additionally suffer from multi-path interference. In addition, the homodyne demodulation is strongly affected by direct-current (DC) offset, flicker noise, and inner mutual coupling [6,10]. On the other hand, the heterodyne technique, using in the demodulation stage a coherent shifted frequency with respect to the carrier [low-intermediate frequency (Low IF)], eliminates DC offset and flicker noise drawbacks but still does not resolve negative effects due to environmental cluttering and internal mutual coupling [6,10]. Recently, improved versions of Doppler radar sensors for millimeter displacement detection, in which the carrier signal is frequency modulated by a sinusoidal wave, have been proposed in [11,12]. In [11], a sinusoidal FM modulation of the optical frequency carrier is used to generate a multi-harmonic spectrum in the received signal so as to exploit the first and second harmonics to extract micro movements of the target. A different solution for estimating heart rate was proposed in [12]. In particular, a sinusoidal frequency modulation is adopted to produce, in conjunction with the chest motion, two secondary harmonics around the Low-IF of the received signal. Both of these improved solutions, working only on the first harmonics, do not allow the adoption of range target discrimination, which is useful in the working scenarios for biomedical applications.

In the context of human environmental localization and vital signs monitoring, FMCW techniques represent the best sensor solutions in terms of range detection, accuracy, and robustness against clutter and multi-path phenomena, even though they are more complex than DCW solutions in terms of architecture design and signal processing [6]. FMCW radar sensors take advantage of the inherent heterodyne demodulation created by the delay that affects the received signal when a frequency modulation is adopted for the transmitted signal. Due to its good performance for targets far away from the FMCW radar, linear modulation (chirp) is the most widely used frequency modulation for this type of radar sensor [6]. The widespread use of these radar systems has prompted scientists and engineers to develop standard hardware architectures to mitigate the cost of the final device. For this purpose, FMCW radars used for vital activity monitoring are typically based on integrated RF components [13,14,15] or on commercial boards [16,17,18,19,20,21,22,23,24,25]. In [13], a 5.8-GHz radar transmission chain has been implemented by assembling together a voltage-controlled oscillator (VCO), a power amplifier, and a power divider, whereas the reception chain is formed by a low noise amplifier, a mixer, and an analog-to-digital converter (ADC). This radar was able to monitor the cardiorespiratory activity of a subject in different chest orientations (front, side, and back). The radars in [14,15] were implemented with off-the-shelf components, and the VCO control was driven using an ad hoc sawtooth generator. These radars are able to distinguish human targets from other objects and accurately measure the subject’s vital signs. For these applications, compact, high-gain antennas, such as patch arrays [8,13], horns [26], or dielectric resonator antennas equipped with dielectric lenses [27], may be usefully employed.

Regarding commercial boards, Infineon produces Silicon Germanium MMIC transceivers operating at 24 GHz in the version with one receiver (BGT24MTR11) [16], two receivers (BGT24MTR12) [17], and two transmitters and two receivers (BGT24LTR22) [18]. On the other hand, Analog Devices provides EV-RADAR-MMIC2 and EV-TINYRAD boards featuring the ADF5901 transmitter, the ADF5904 receiver, the ADF4159 phase-locked loop, all in 24 GHz MMIC technology [19], and a multiple-input multiple-output (MIMO) integrated antenna array [20]. Using a board equipped with the BGT24MTR11 chipset and external horn antennas, it has been demonstrated that, with a single FMCW radar, both respiratory activity and position can be measured simultaneously [21,22]. In particular, respiratory activity can be reconstructed by analyzing the phase variations in the harmonic related to the target position. In [23], a new technique to remove radar motion effects for accurate vital signs detection has been presented and experimentally demonstrated using a BGT24MTR11 transceiver board equipped with transmitting and receiving antennas. In [24], the Platypus system based on the EV-TINYRAD board was proposed for the simultaneous tracking of micro-displacements of multiple objects from a single observation point. The same board was used to investigate the effect of various radar parameters such as bandwidth, power, antennas, and signal processing techniques on the performance of remote vital signs monitoring [25].

In all the considered radars, the VCO chipset is driven by ramp or triangular signals to generate the chirp to be transmitted. These signals are produced by a low-frequency signal generator, a data acquisition (DAQ) board, or a phase-locked loop (PLL) system. This latter solution is the most accurate since the chirp frequencies are generated in discrete steps and phase-locked to an extremely stable reference oscillator. However, when using a ramp signal, a sharp frequency jump is introduced between the end and the beginning of the ramp, which leads to distortion in the signal frequency spectrum [28,29]. This problem does not occur if the VCO is driven by a triangular wave. On the other hand, a trigger signal has to be generated at each slope change to acquire a coherent IF signal output. When the VCO is controlled by a ramp wave, the trigger signal allows for sampling the received signal in a unique and coherent manner. Conversely, when the control occurs with a triangular wave, there are two transitions that create an ambiguity in the acquired signal. Furthermore, due to internal mutual coupling, the detected signals are affected by systematic interference that could mask the useful signal, thus making it difficult to identify the most suitable harmonic bin to perform motion signal processing [30]. This is particularly stringent for objects monitored at a short distance from the radar, where the frequency bands of the internal interference and of the signal reflected by the target overlap.

In this work, a new low-complexity modulation scheme for frequency-modulated continuous-wave (FMCW) radars, which solves the above-mentioned problems, is presented. The proposed scheme is free from the synchronization constraint and exploits a sine wave as the VCO drive signal, giving rise to a sinusoidal frequency-modulated continuous-wave (SFMCW) RF signal. Furthermore, the adopted modulation scheme makes the monitoring system much less sensitive to internal mutual coupling even where the monitored objects are close to the radar, as is the case of heart and respiratory rate monitoring. This article is organized into five sections. The considered modulation is described theoretically in Section 2; the experimental implementation of the radar, improved using a PLL board, is reported in Section 3; experimental measurements to verify the system’s ability to monitor millimetric displacements, highlighting its robustness against internal mutual coupling, are described in Section 4; remarks and conclusions are reported in Section 5.

## 2. Materials and Methods

### 2.1. Background

A typical schematic of a canonical chirp-type FMCW radar system, based on a linear frequency profile modulation employing a sawtooth wave as the modulating signal, is reported in Figure 1. This hardware architecture raises some important issues to be considered during the design phase in order to obtain an accurate measurement device; a first important point concerns the generation of an RF signal that is linearly modulated in frequency, while an equally fundamental issue refers to the synchronization of the I/Q output signal acquisition with the ramp pattern of FM modulation, so as to avoid acquiring the signal during the time interval between the beginning of the modulating ramp and the end of the return signal step [see Figure 1b]. No less important is the impact of antenna mismatch and internal mutual coupling interference on detected signals, which results in an additional challenging design goal.

As shown in Figure 1a, an FMCW radar consists of a signal generator that supplies a sawtooth-like low-frequency (LF) modulating signal with period T, which, via a VCO, generates a frequency-modulated (FM chirp) RF signal that linearly covers the working frequency band $B=f_{max}-f_{min}$. The transmitted signal passes through the RF circuitry, is radiated by the transmitting antenna, propagates, hits the target, and, delayed by the time of flight, is captured by the receiving antenna. Finally, it is multiplied by a mixer with the transmitted modulated RF signal. The baseband I/Q signals, obtained from the mixer circuit, inherently contain information about the distance of the target in their spectrum. Therefore, by computing the Fourier transform of the complex signal obtained by combining the I/Q outputs, the harmonic ${\hat{f}}_{i}$ at which the spectrum reaches its peak can be used to identify the position of the target, while the phase ϕ of the corresponding spectral component can be used to evaluate small variations in distance with respect to the mean position of the target. By analytically modeling the individual blocks of the radar system and assuming a sawtooth-like modulating signal, the following relations which link the baseband received signal $s_{R}\left(t\right)$ to the overall delay time τ [1] and to the mutual coupling interference [30] can be derived:(1)$$\begin{array}{c} s_{R}\left(t\right)=A_{R}e^{-j\left[\Phi \left(t-\tau \right)-\Phi \left(t\right)+\varphi_{0}^{\prime }\right]}+A_{MC}e^{-j\left[\Phi \left(t-\tau_{MC}\right)-\Phi \left(t\right)+{\varphi_{0}^{\prime }}_{MC}\right]}≅\\ \;≅A_{R}e^{j\left[2\pi f_{i}\left(t\right)\tau -\varphi_{0}^{\prime }\right]}+A_{MC}e^{j\left[2\pi f_{i}\left(t\right)\tau_{MC}-{\varphi_{0}^{\prime }}_{MC}\right]}=\\ ≅\left\{\begin{array}{l} \begin{array}{c} {A_{R}\;e}^{j\left[2\pi f_{o}\tau +\pi B\tau \;\left(-1+\frac{2t}{T_{r}}\right)-\varphi_{0}^{\prime }\right]}\\ +A_{MC}e^{j\left[2\pi f_{o}\tau_{MC}+\pi B\tau_{MC}\;\left(-1+\frac{2t}{T_{r}}\right)-{\varphi_{0}^{\prime }}_{MC}\right]} \end{array}\;\;\;\;\;\;\;\;\;\;\;\;\;\;0\leq t<T_{r}\\ \begin{array}{c} A_{R}e^{j\left[2\pi f_{o}\tau -\pi B\tau -\varphi_{0}^{\prime }\right]}+\\ +A_{MC}e^{j\left[2\pi f_{o}\tau_{MC}-\pi B\tau_{MC}-{\varphi_{0}^{\prime }}_{MC}\right]} \end{array}\;\;\;\;\;\;\;\;\;\;\;\;\;\;\;\;\;\;\;\;\;\;\;\;\;\;\;T_{r}\leq t<T=T_{r}+T_{idle} \end{array}\right. \end{array}$$
where(2)$$\Phi \left(t\right)=\Phi_{0}+2\pi \int_{0}^{t}f_{i}\left(\xi \right)d\xi =\Phi_{0}+2\pi \int_{0}^{t}\left[f_{o}+\frac{B}{2}\;\left(-1+\frac{2\xi }{T_{r}}\right)\right]d\xi$$

In (1) and (2), $A_{R}$ and $A_{MC}$ denote the amplitude of the received and the interfering mutual coupling signal, respectively, $\varphi_{0}^{\prime }$ and ${\varphi_{0}^{\prime }}_{MC}$ identify spurious phase shifts, $T_{r}$ indicates the time taken by the ramp of the modulating signal to complete a scan cycle from $f_{min}$ to $f_{max}$, $f_{o}$ is the band central frequency, while $T_{idle}$ is the waiting time between successive ramps employed to make the transient effects vanish. In (1), $\tau_{MC}$ indicates the total delay time of the mutual coupling signal due to the internal path. It must be noted that $\tau_{MC}\ll \tau$, since the internal path is much shorter than the round-trip experimented by the radiated signal.

If the following condition is satisfied(3)$$1\ll B\tau \ll T_{r}$$
the mutual coupling interfering signal frequency band is far from that of the useful signal, and therefore, after synchronization and low-pass filtering, Equation (1) can be rewritten in its canonical form:(4)$$s_{R}\left(t\right)=A_{R}e^{j\left[\varphi \left(t\right)+\phi +\varphi_{0}^{\prime \prime }\right]}$$
with(5)$$\varphi \left(t\right)=2\pi \frac{B}{T_{r}}\tau \;t\;\;\;\;\;\Rightarrow \;\;\;\;\;{\hat{f}}_{i}=\frac{1}{2\pi }\frac{d\varphi \left(t\right)}{dt}=\frac{B}{T_{r}}\tau$$(6)$$\phi =2\pi \left(f_{o}-\frac{B}{2}\right)\;\tau =2\pi f_{min}\;\tau \;\;\;$$(7)$$\varphi_{0}^{\prime \prime }=-\varphi_{0}^{\prime }.$$

Under condition (3), satisfied if the monitored object is far from the radar, it follows that [see (5) and (6)] the instantaneous frequency ${\hat{f}}_{i}$ and phase term ϕ depend linearly on the time of flight τ but with a different proportionality coefficient (phase-delay sensitivities).

All FMCW radar systems with chirp modulation exploit these relationships for monitoring vital parameters, using (5) to identify the subject’s position (to discriminate the subject from other obstacles/subjects) and (6) to monitor cardiac and respiratory activities. The adoption of the sawtooth-like wave chirp has become a sort of standard in FMCW radar systems, so much so that commercial PLL devices [19,20] contain specific firmware for digital step generation of the sawtooth-like modulation wave. In addition to the problem of accurately generating linearly frequency-modulated RF signals, another important issue is the correct synchronization of the start of the acquisition process within each chirp period of the signal detected by the mixer circuit. As indicated in Figure 1b, the signal acquisition has to start considering the time of flight in such a way as to exclude from the acquisition the signal during the back-off time (time between the beginning of the modulating ramp and the end of the return signal ramp) and the idle time ($T_{idle}$) used for the transient effects to vanish. This careful timing avoids detecting incorrect signals coming out of the mixer. This problem is especially significant when adopting high monitoring rates (short $T_{r}$), since the right-hand side of (3) does not hold anymore. Finally, as concerns the mutual coupling interference, its masking effect on the useful signal can be evaluated by computing the Fourier transform of (1) in the rise time interval ($0\leq t\leq T_{r}$)(8)$$\frac{1}{T_{r}}\int_{0}^{T_{r}}s_{R}\left(t^{\prime }\right){\;e}^{-j2\pi ft^{\prime }}dt^{\prime }={\;A_{R}e}^{j\left(-\pi {fT}_{r}+2\pi \;f_{o}\tau -\varphi_{0}^{\prime }\right)}\frac{sin\left[\pi \left(f\;T_{r}-B\tau \right)\right]}{\pi \left(f\;T_{r}-B\tau \right)}+\;+{\;A_{MC}e}^{j\left(-\pi {fT}_{r}+2\pi \;f_{o}\tau_{MC}-\varphi_{0_{MC}}^{\prime }\right)}\frac{sin\left[\pi \left(f\;T_{r}-B\tau_{MC}\right)\right]}{\pi \left(f\;T_{r}-B\tau_{MC}\right)}.$$

The peak of the magnitude of the first term on the right-hand side of (8) is given by ${\hat{f}}_{i}$ reported in (5); at this frequency the magnitude of the interference spectrum [second term on the right-hand side of (8)], considering $\tau_{MC}\ll \tau$, can be maximized by the following expression(9)$$\left|A_{MC}\right|\frac{\left|sin\left[\pi \;B\left(\tau -\tau_{MC}\right)\right]\right|}{\pi \;B\left(\tau -\tau_{MC}\right)}≅\;\left|A_{MC}\right|\frac{\left|sin\left(\pi \;B\;\tau \right)\right|}{\pi \;B\;\tau }\leq \frac{\left|A_{MC}\right|}{\pi \;B\;\tau }.$$

Since the amplitude of the mutual coupling signal $\left|A_{MC}\right|$ generally is much greater than the received signal $\left|A_{R}\right|$, even in the case of the target not far from the radar [30], when Bτ is not much greater than 1 [condition on the left-hand side of (3) is not met], the spectrum of the interference signal masks the useful received signal, with inevitable consequences on the detection of phase (6). Of course, if the synchronization fails to capture the most significant part of the received signal, the phase detection (6) will get worse.

To this end, a novel FMCW radar scheme that is free from the above-mentioned operational constraints and, therefore, allows the acquisition of the detected signal even in a non-synchronized manner, with high robustness with respect to the mutual coupling interfering signal, is proposed in this paper. This technical solution reduces the design complexity of the radar without worsening its performance in terms of accuracy and application areas.

### 2.2. Sinusoidal Frequency-Modulated Continuous Wave (SFMCW) Radar System

The proposed FMCW radar scheme exploits a sine wave as a modulating signal. The equation of a sinusoidally FM-modulated signal with carrier frequency $f_{o}$, modulating frequency $f_{m}$, and modulation index $m=\frac{B}{2\;f_{m}}\gg 1$ assumes the following form:(10)$$s_{TX}\left(t\right)=A_{TX}\;cos\left(2\pi f_{o}t+\frac{B}{2\;f_{m}}\sin2\pi f_{m}t\right)$$
where the parameter $A_{TX}$ denotes the amplitude of the modulated signal, and B is the frequency band “spanned” by the instantaneous frequency $f_{i}$ of the modulated signal that covers the interval $\left[f_{o}-\frac{B}{2\;},f_{o}+\frac{B}{2\;}\right]$:(11)$$f_{i}=f_{o}+\frac{B}{2\;}\cos2\pi f_{m}t$$

The band B practically coincides with the frequency-modulated signal bandwidth, which can be calculated using Carson’s rule [31]:(12)$$B_{FM}≅2f_{m}(1+m)=2f_{m}(1+\frac{B}{2\;f_{m}})≅B$$

The received signal $s_{RX}\left(t\right)$, delayed by the time of flight τ with respect to the generated signal, can be expressed as(13)$$s_{RX}\left(t\right)=A_{RX}\;cos\left(2\pi f_{o}\left(t-\tau \right)+\frac{B}{2\;f_{m}}\sin\left(2\pi f_{m}\left(t-\tau \right)\right)+\phi_{o}\right)$$
where $A_{TX}$ identifies the amplitude of the received signal, and $\phi_{o}$ denotes the additional phase shift due to the scattering process caused by the target and that caused by the RF circuitry. In this context, the phase $\phi_{o}$, very slowly variable over time, can be considered substantially independent of time.

Using a mixer with a proportionality constant $k_{m}$, the $s_{I}\left(t\right)$ (in-phase) and $s_{Q}\left(t\right)$ (in-quadrature) received signals, after filtering, take the following forms:(14)$$\begin{array}{l} s_{I}\left(t\right)=k_{m}s_{RX}\left(t\right)cos\left(2\pi f_{o}t+\frac{B}{2\;f_{m}}\sin2\pi f_{m}t\right)=\\ \;\;\;\;\;\;\;\;\;\;=\frac{k_{m}A_{RX}}{2}\cos\left(-2\pi f_{o}\tau +\frac{B}{2\;f_{m}}\sin\left(2\pi f_{m}\left(t-\tau \right)\right)-\frac{B}{2\;f_{m}}\sin\left(2\pi f_{m}t\right)+\phi_{o}\right) \end{array}$$(15)$$\begin{array}{l} s_{Q}\left(t\right)=k_{m}s_{RX}\left(t\right)sin\left(2\pi f_{o}t+\frac{B}{2\;f_{m}}\sin2\pi f_{m}t\right)=\\ \;\;\;\;\;\;\;\;\;\;\;=\frac{k_{m}A_{RX}}{2}\sin\left(-2\pi f_{o}\tau +\frac{B}{2\;f_{m}}\sin\left(2\pi f_{m}\left(t-\tau \right)\right)-\frac{B}{2\;f_{m}}\sin\left(2\pi f_{m}t\right)+\phi_{o}\right) \end{array}$$

Observing that ${2\pi f}_{m}\tau \ll 1$, (14) and (15) can be rewritten as follows:(16)$$s_{I}\left(t\right)=\frac{{k_{m}A}_{RX}}{2}\cos\left(2\pi f_{o}\tau +\pi B\tau \cos\left(2\pi f_{m}t\right)-\phi_{o}\right)$$(17)$$s_{Q}\left(t\right)=\frac{k_{m}A_{RX}}{2}\sin\left(2\pi f_{o}\tau +\pi B\tau \cos\left(2\pi f_{m}t\right)-\phi_{o}\right)$$

Since the signals $s_{I}\left(t\right)$ and $s_{Q}\left(t\right)$ are periodic with the same period as the modulating signal $T_{m}=1/f_{m}$, their analytical representation in terms of the Fourier series can be written as follows:(18)$$\begin{array}{l} s_{I}\left(t\right)=\frac{k_{m}A_{RX}}{2}\left[\cos\left(2\pi f_{o}\tau -\phi_{o}\right)J_{0}\left(\pi B\tau \right)+\right.\\ \;\;\;\;\;\;\;\;\;\;+\sum_{q=1}^{\infty }2{\left(-1\right)}^{q}\cos\left(2\pi f_{o}\tau -\phi_{o}\right)J_{2q}\left(\pi B\tau \right)\cos\left(4q\pi f_{m}t\right)+\\ \;\;\;\;\;\;\;\;\;\;\;-\sum_{q=0}^{\infty }2{\left(-1\right)}^{q}\sin\left(2\pi f_{o}\tau -\phi_{o}\right)J_{2q+1}\left(\pi B\tau \right)\cos\left(2(2q+1)\pi f_{m}t\right)]\;\; \end{array}$$(19)$$\begin{array}{l} s_{Q}\left(t\right)=\frac{k_{m}A_{RX}}{2}\left[\sin\left(2\pi f_{o}\tau -\phi_{o}\right)J_{0}\left(\pi B\tau \right)+\right.\\ \;\;\;\;\;\;\;\;+\sum_{q=1}^{\infty }2{\left(-1\right)}^{q}\sin\left(2\pi f_{o}\tau -\phi_{o}\right)J_{2q}\left(\pi B\tau \right)\cos\left(4q\pi f_{m}t\right)+\\ \;\;\;\;\;\;\;\;\;\;+\sum_{q=0}^{\infty }2{\left(-1\right)}^{q}\cos\left(2\pi f_{o}\tau -\phi_{o}\right)J_{2q+1}\left(\pi B\tau \right)\cos\left(2(2q+1)\pi f_{m}t\right)]\; \end{array}$$

Combining $s_{I}\left(t\right)$ and $s_{Q}\left(t\right)$, the complex signal $s_{c}\left(t\right)=s_{I}\left(t\right)+js_{Q}\left(t\right)$ can be expressed as follows:(20)$$\begin{array}{l} s_{c}\left(t\right)=\frac{k_{m}A_{RX}}{2}e^{j2\pi f_{o}\tau -j\phi_{o}}\times \\ \;\;\;\;\;\;\;\;\;\;\times [J_{0}\left(\pi B\tau \right)+\sum_{q=1}^{\infty }2{\left(-1\right)}^{q}J_{2q}\left(\pi B\tau \right)\cos\left(4q\pi f_{m}t\right)+\\ \;\;\;\;\;\;\;\;\;\;+j\sum_{q=0}^{\infty }2{\left(-1\right)}^{q}J_{2q+1}\left(\pi B\tau \right)\cos\left(2(2q+1)\pi f_{m}t\right)]\; \end{array}$$

From (20), it can be observed that the time delay τ appears both in the phase term of the exponential and in the argument of the Bessel functions. Similar to the case of the sawtooth chirp [see Equation (6) with $f_{o}$ instead of $f_{min}$], the phase term of the exponential can be used to evaluate vital parameters such as respiratory and cardiac rhythm, while the one present in the argument of the Bessel functions can be used to identify the subject’s position. In addition, in the absence of modulation $\left(B=0\right)$ or when the argument of the Bessel functions tends to zero (target very close to the radar), the complex signal in (20) coincides with that obtained with DCW radars, since within the square brackets only the Bessel function of order zero remains, assuming value 1. The above observation shows that when the target is very close to the radar, the SFMCW radar behavior becomes equivalent to that of the homodyne technique, with the inherent DC offset and flicker noise problems. Therefore, under these conditions, the best solution is the one based on the Low-IF CW heterodyne technique [6].

From (19), the following Fourier transform coefficients of the complex signals $s_{c}\left(t\right)$ are easily identified(21)$$\;{I\left\{s_{c}\left(t\right)\right\}}_{p}=\frac{k_{m}A_{RX}}{2}e^{j2\pi f_{o}\tau -j\phi_{o}}\left\{\begin{array}{l} J_{0}\left(\pi B\tau \right)\;\;\;\;\;\;\;\;\;\;\;\;\;\;\;\;\;\;\;\;\;\;\;\;\;\;\;\;\;p=0\\ {\left(-1\right)}^{\frac{p}{2}}J_{p}\left(\pi B\tau \right)\;\;\;\;\;\;\;\;\;\;\;\;\;\;\;\;\;p\;even\\ j{\left(-1\right)}^{\frac{p-1}{2}}J_{p}\left(\pi B\tau \right)\;\;\;\;\;\;\;\;\;p\;odd \end{array}\right.$$

Therefore, indicating with $\hat{p}$ the harmonic order corresponding to the maximum coefficient magnitude, the phase variation term (6), with $f_{o}$ instead of $f_{min}$, can be extracted from the phase of this coefficient. According to the analytical properties of the Bessel functions [32], the peak amplitude of $J_{p}\left(\pi B\tau \right)$, under the condition πBτ≫1 is reached for $\hat{p}≅\pi B\tau$.

Regarding the impact of the mutual coupling interfering signal, the terms in (21) corresponding to the peak harmonic $\hat{p}≅\pi B\tau$ of the received signal can be approximated by applying the small argument Bessel function approximation [32] as follows:(22)$${I\left\{s_{c_{MC}}\left(t\right)\right\}}_{\hat{p}}≅\frac{\left|k_{m}A_{MC}\right|}{2\hat{p}!}{\left(\frac{\pi B\tau_{MC}}{2}\right)}^{\hat{p}}.$$

Equation (22) implies, as a direct consequence of sinusoidal modulation (11), that only the term p=1 makes a significant contribution, while the other terms are negligible. Therefore, only the fundamental harmonic is affected by the mutual coupling signal, so the proposed modulation scheme is very robust against intrinsic internal interference.

## 3. Results

### 3.1. Numerical Results

#### 3.1.1. Effect of Mutual Coupling Interference and Synchronization

To validate the proposed modulation scheme, a numerical test, in which the received signal is generated by an oscillating target representative of a heartbeat with a rate $f_{H}=1\;Hz$ (60 bpm) and a peak displacement $r_{H}=1\;mm$, is presented. The received signal is characterized by the following parameters $f_{o}=24.125\;GHz$, τ=10 ns (corresponding to an equivalent “in air” distance from the target at rest equal to $r_{0}=1.5\;m$), B=250 MHz, $f_{m}=50\;Hz$, $k_{m}A_{RX}=0.4\;V$, and $\phi_{o}=0$, while the interference signal is characterized by $\tau_{MC}=1\;ns$, $k_{m}A_{MC}=2.0\;V$, and $\phi_{o_{MC}}=0$ (as pointed out in [30], $A_{MC}$ larger than $A_{RX}$ is set). Considering the target stationary during the acquisition period, when the delay time is τ=10 ns, the signals $s_{I}\left(t\right)$ and $s_{Q}\left(t\right)$ exhibit the time behavior shown in Figure 2.

The spectrum amplitude of the complex signal $s_{c}\left(t\right)$ is shown in Figure 3. Specifically, Figure 3a shows both the spectra of the received signal and interference signal to highlight the corresponding harmonic amplitudes, while Figure 3b shows the overall spectrum of $s_{c}\left(t\right)$. As expected, the interference signal is actually limited to the first harmonic, and therefore the peak harmonic of $s_{c}\left(t\right)$, which appears at index $\hat{p}=6$, is not affected by the interference signal, thus preserving the information about the position and motion of the monitored object.

For comparison purposes, the same case is now analyzed by considering the canonical chirp modulation. Referring to Figure 1, the sawtooth-like signal is characterized by a ramp rise time $T_{r}$ equal to 80% of the period ($T_{r}=16\;ms$), and the remaining 20% is used as idle time to recover from any transitory effect. The signals $s_{I}\left(t\right)$ and $s_{Q}\left(t\right)$, based on chirp modulation, exhibit the time behavior shown in Figure 4. In Figure 5, the corresponding spectra are shown.

From Figure 5, it appears clearly that the interference signal significantly affects the amplitude of harmonics p=2 and p=3, where the largest amplitudes of the received signal spectrum are present. An even worse situation occurs if the synchronization for the acquisition of the received signal is not achieved. Figure 6 shows the spectrum of the overall signal $s_{c}\left(t\right)$ when a misalignment in the acquisition time windowing of 10% of the ramp rise time $T_{r}$ occurs (absence of synchronization).

As it appears from Figure 6, the incorrect synchronization introduces a fatal perturbation of the spectrum so that the peaks on the *p* = 2 and *p* = 3 harmonics are totally masked. Of course, adopting the sinusoidal modulation scheme does not expose the radar system to this kind of drawback, demonstrating how the proposed modulation scheme is more robust than the canonical chirp modulation technique in short-range acquisition.

#### 3.1.2. Motion Detection

As concern the radar behavior with respect to motion detection, the following oscillating movement of the target around the resting position $r_{0}$,(23)$$r\left(t\right)=r_{0}+r_{H}\sin\left(2\pi f_{H}t\right)$$
can be considered. The comparison between the time behavior of the object displacement (23) and the movement extracted using the phase variation term according to the proposed modulation scheme [see Equation (21)] is reported in Figure 7. As it can be observed, the proposed model is able to evaluate the millimetric movement of the object accurately.

### 3.2. Experimental Results

To point out the validity of the proposed modulation scheme in realistic scenarios, several experimental tests were performed. For this purpose, an SFMCW radar system was designed using an Infineon BGT24MTR11 [24] module integrated with an ADF4159 PLL board [19] and two Pasternack horn antennas employed as transmitting and receiving antennas, respectively. As shown in Figure 8, the entire system is controlled by an NI PXI 1042 device equipped with a NI-5411 waveform generator and an 8-port NI-4472 analog to digital acquisition board, managed through a LabVIEW program.

To force the PLL board to generate a 50 Hz sinusoidal modulation signal for the VCO input of the Infineon BGT24MTR11 module, a periodic triggering signal is generated by the NI-5411 waveform generator with a rate of 1 MHz to feed the external trigger port of the PLL. A pulse train was used with a pulse duration of 1 μs, located according to the following sequence for each half-period $T_{H}=10\;ms$:(24)$$t_{k}=\frac{T_{H}}{2\pi }\left[arcsin\left(\frac{2}{N_{st}-1}k-1\right)+arcsin\left(\frac{2}{N_{st}-1}\left(k+1\right)-1\right)\right]+\frac{T_{H}}{2}\;\;\;\;\;\;\;0\leq k\leq N_{st}-2$$

The ADF4159 PLL board generates a staircase frequency modulation with a sinusoidal shape with $N_{st}=1000$ steps. Figure 9 reports the VCO signal sent as input to the Infineon BGT24MTR11, showing the sinusoidal modulation provided by the PLL board and the corresponding spectrum of the RF modulated signal, measured after a divide-by-16 prescaler module. As reported in Figure 9, the SFMCW signal occupies, with very sharp edges, a frequency band from 24.000 GHz to 24.250 GHz. Consequently, from (6) and (21), the phase-range sensitivity of the radar is $S_{r}=\frac{1}{c_{0}}\frac{d\;\phi }{d\tau }=\frac{4\pi \;f_{o}}{c_{0}}=1.01\;rad/mm$.

#### 3.2.1. Mutual Coupling Interference

The designed SFMCW radar system has been employed to verify the validity of the proposed modulation scheme. Although several tests have been performed, only one is reported here for the sake of brevity. In particular, in this test, the measurement of a millimetric damped oscillation of a metallic panel of size 20 cm×30 cm, located about 1.0 m far from the radar antennas, was performed. The reference displacement was detected using the analog output laser sensor LMI LDS-80-10, with a range sensitivity of 1 V/mm. Figure 10 shows an example of $s_{I}\left(t\right)$ and $s_{Q}\left(t\right)$ signals captured by the radar, while Figure 11 reports the corresponding spectrum, from which the highest Fourier coefficient at the fifth harmonic can be observed. From these figures, it is evident that the strong reflection of the nearby metal panel makes the amplitude of the mutually coupled interference signal lower than the received useful signal. For less scattering objects, such as the human body, this advantageous condition does not occur, as it appears from the analysis of Figure 12, Figure 13, Figure 14 and Figure 15, which refer to the case when the metal panel is replaced by a metal sphere with a diameter of 30 cm, with its center at 1 m far from the radar antennas (same location of the metal plane). Since the sphere scatters less electromagnetic energy than the metal panel, the mutual coupling interference, which does not depend on the received signals, affects the overall acquired signal more significantly.

Consequently, the corresponding spectra show that while for the SFMCW scheme, the major harmonic components are not affected by the coupling interference signal (peak index equal to 5, due to an overall τ≅9 ns, since the portion of sphere surface illuminated by the radar is closer to the antennas than the metal panel), for chirp modulation it is evident how the significant harmonic [$\hat{p}=2≅B\tau$ in this case (see Equation (8)] is masked by the disturbance harmonic contribution.

#### 3.2.2. Motion Detection

Finally, regarding the monitoring of a target in motion, Figure 16 shows a comparison between the panel motion measured by the SFMCW radar (a) and by the laser sensor (b) (the metal sphere was not monitored in motion due to the difficulty of easily producing a significant oscillatory displacement). By comparing the two curves, a good agreement can be appreciated. In particular, the complete synchronization between the two measured movements and the amplitude of the detected sub-millimeter displacement can be observed. The main cause of the discrepancies between the two time behaviors is the differences between the two different types of measurements taken. The SFMCW radar measures the surface average movement of the panel, as the beam width of the radar antennas covers the entire panel; meanwhile, the laser sensor measures the local displacement of the panel on a defined spot, which can also be characterized by high-order subdominant motion modes. In any case, the overall mean relative error between the two signals is about 0.1 mm, validating the performance of the proposed technique.

## 4. Discussion

As discussed and highlighted in the previous sections, the key design elements of FMCW radars, which have a great impact on radar performance as well as on hardware complexity and stringent processing requirements, are the VCO drivers, RF mixing circuits, and synchronization procedures. VCOs and RF mixers, made with MMIC components [15,33], packaged components [13,34], or silicon germanium integrated circuits [16,22], are the main elements of the radar transmitting and receiving channel (see Figure 1). To generate the frequency sweep, the VCO is driven with a highly integrated fractional-N frequency synthesizer (PLL) [35] or, for cheaper and more portable solutions, with integrated circuits based on operational amplifiers or transistors [15]. In the second case, circuits based on Miller integrators [15] or quartz-based sinusoidal oscillators [36] can be used to generate the required modulation, with the latter (sinusoidal quartz oscillators for SFMCW radars) being very reliable although very inexpensive.

Moreover, as discussed in Section 2, an important requirement in the design of FMCW radars is the isolation between ports of the RF mixer [37]. Poor port isolation introduces high levels of signal interference, worsening the radar performance in detecting nearby targets, such as in applications concerning heart and respiratory rate monitoring. Therefore, high-cost RF mixers are required for reliable FMCW radar solutions, while for SFMCW radars, the high robustness with respect to interference signals allows the use of cheaper RF mixer circuits.

Finally, the total absence of a synchronization procedure for SFMCW radars concerning the start and stop of the acquisition process within each chirp period of the signal detected by the mixing circuit makes SFMCW radars a valid alternative to standard techniques, such as DCW, Low-IF CW, and chirp FMCW, for accurate measurements of sub-millimeter displacements compatible with breath and heart movements. Moreover, maintaining the ability to discriminate the monitored subject by choosing the corresponding harmonic coefficient, as in the canonical linear chirp FMCW modulation scheme, makes confident that the proposed SFMCW radars can be effectively adopted for breath and heart rate monitoring. Additionally, since spectrum harmonics close to the highest one have significant levels and phase coherence, they can be efficiently adopted to improve the signal-to-noise ratio (SNR) of the motion signal by applying the slow-time correlation method developed for monitoring target displacement that produces multiple coherent reflections [38].

With regard to human testing, an analysis of compliance of the radiated electromagnetic field with safety standards on human exposure [39] was carried out. In particular, considering the maximum output power by the radar circuit (15 dBm), the gain of the employed PE985212F-15 horn antennas by Pasternack (15 dBi), the radiated electromagnetic power density 1 m far from the antenna is about $0.08\;W/m^{2}$ which is about 125 times lower than the maximum human body EM exposure limits proposed in ICNIRP guidelines ($10\;W/m^{2}$ at 24 GHz) [39]. This exposure estimation shows that the developed SFMCW radar can be used to monitor breath and heart rate without any risk to volunteer subjects and patients. The authors are currently awaiting ethics approval from the Sapienza University of Rome Ethics Committee for Transdisciplinary Research to begin testing on volunteer human subjects.

## 5. Conclusions

A new modulation scheme for frequency-modulated continuous-wave (FMCW) radars suitable for millimeter target displacement detection was presented. A 24 GHz radar consisting of an Infineon BGT24MTR11 module equipped with a PLL board and horn antennas, controlled by a LabVIEW program by means of a generator waveform and an acquisition board installed in a PXI chassis, was developed to validate the proposed modulation scheme. As demonstrated by the analytical model and experimental results, the proposed scheme is free from any synchronization constraints and robust against internal mutual coupling, making it very effective in reducing the design complexity of the radar without worsening its performance in terms of accuracy and operating ranges. Numerical and experimental results obtained with the designed 24 GHz radar have shown the suitability of the proposed modulation scheme for monitoring very small movements of targets, with levels of accuracy compatible with laser sensors and thus consistent with those typically required to detect basic vital activities (heartbeat and respiration). This low-complexity modulation scheme can be widely adopted in the future to make very inexpensive but high-performance monitoring radars operating even at higher frequencies (60 GHz and above), where hardware simplification is a key element in their design.

## Figures and Tables

**Figure 1 sensors-25-02765-f001:**
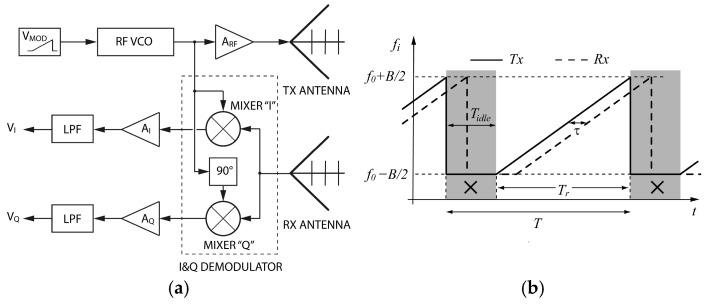
Schematic of an FMCW radar with quadrature demodulation (**a**); time behavior of the transmitted (solid line) and received (dashed line) frequencies (**b**). The interval to be avoided in sampling the received signal is highlighted in (**b**).

**Figure 2 sensors-25-02765-f002:**
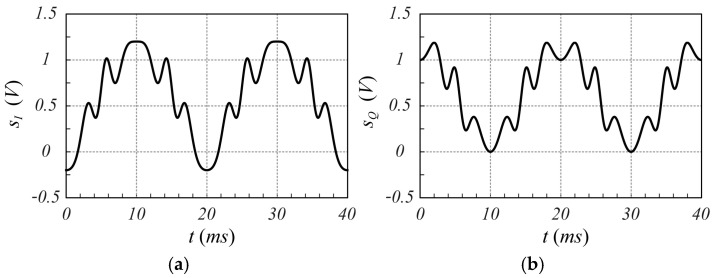
Time behavior of the signals (**a**) $s_{I}\left(t\right)$ and (**b**) $s_{Q}\left(t\right)$, evaluated using (16) and (17) taking into account both the received signal and the interference signal under the hypothesis of sinusoidal modulation.

**Figure 3 sensors-25-02765-f003:**
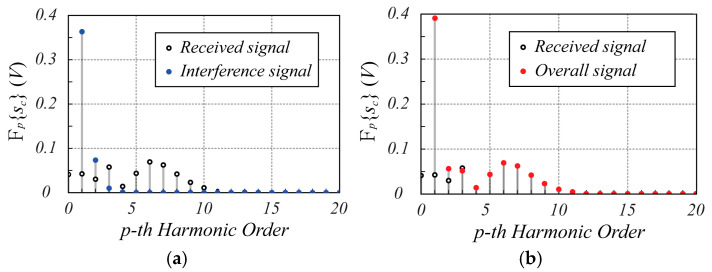
The spectrum magnitude of the $s_{c}\left(t\right)$ signal calculated using (21). In (**a**,**b**), the black dots indicate the received signal, the blue dots indicate the interference signal, while the red dots in (**b**) refer to the overall spectrum of $s_{c}\left(t\right)$. In both graphs, the 0-th harmonic of the interference signal is not reported to improve the spectrum readability since its value is much greater than that of all the other contributions. Note that at harmonic $\hat{p}=6$, where the largest amplitude of the received signal spectrum is present, the interference signal does not affect the useful signal.

**Figure 4 sensors-25-02765-f004:**
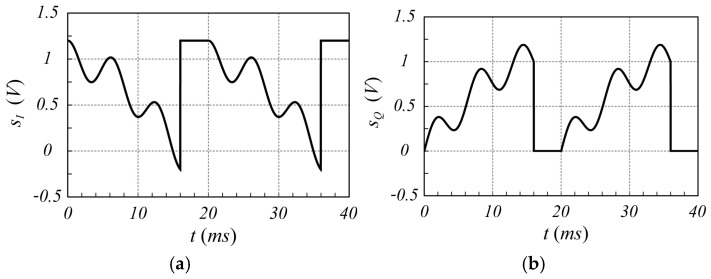
Time behavior of the signals (**a**) $s_{I}\left(t\right)$ and (**b**) $s_{Q}\left(t\right)$ in the case of chirp modulation, evaluated using (1), taking into account both the received and the interference signal.

**Figure 5 sensors-25-02765-f005:**
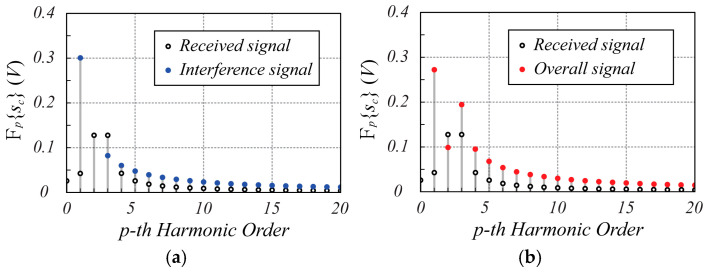
Spectrum magnitude of $s_{c}\left(t\right)$ referring to chirp modulation. In (**a**,**b**), the black dots indicate the received signal, the blue dots indicate the interference, while the red dots in (**b**) refer to the overall spectrum of $s_{c}\left(t\right)$. In both graphs, the 0-th harmonic of the interference signal is not reported to improve the spectrum readability since its value is much greater than that of all the other contributions. Note that at harmonics p=2 and p=3, where the largest amplitudes of the received signal spectrum are present, the interference signal significantly affects the useful signal.

**Figure 6 sensors-25-02765-f006:**
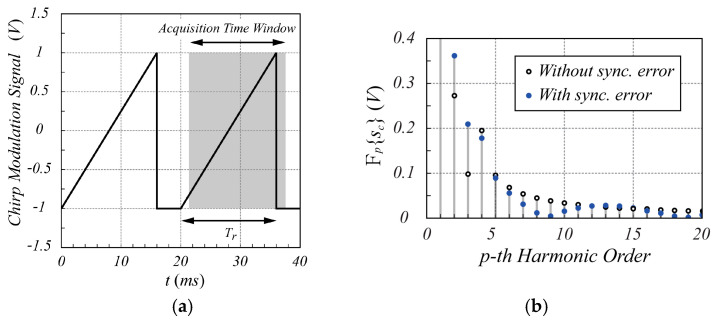
Chirp modulation signal and spectrum magnitude of the time window acquired $s_{c}\left(t\right)$. In (**a**), the gray region identifies the acquisition time window misaligned by 10% of the ramp rise time. The black dots in (**b**) indicate the acquired spectrum without synchronization error, while the blue dots show the spectrum with an incorrect acquisition time window.

**Figure 7 sensors-25-02765-f007:**
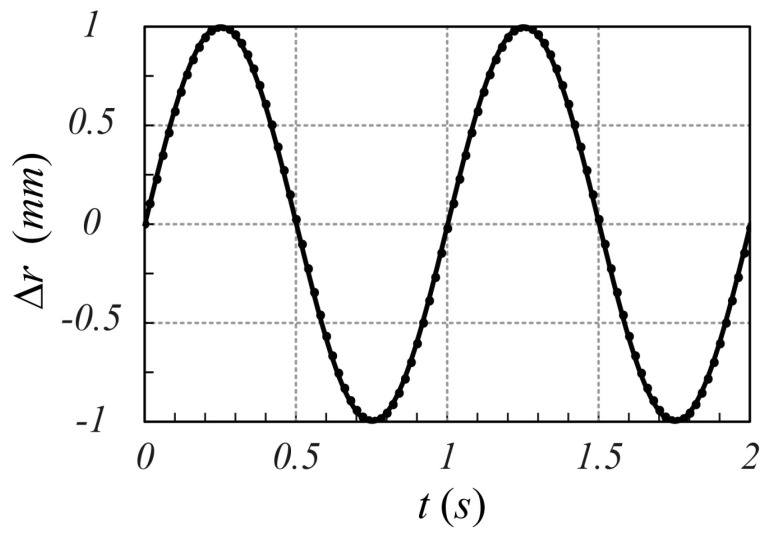
Comparison between the object displacement given by (23) (solid line) and movement extracted by the phase variation term present in (21) (dotted line). Note that the motion sampling rate is 50 Hz.

**Figure 8 sensors-25-02765-f008:**
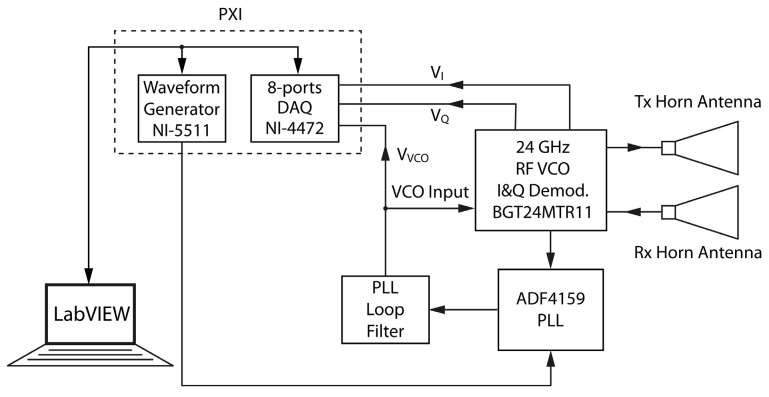
Block diagram of the designed 24-GHz SFMCW radar.

**Figure 9 sensors-25-02765-f009:**
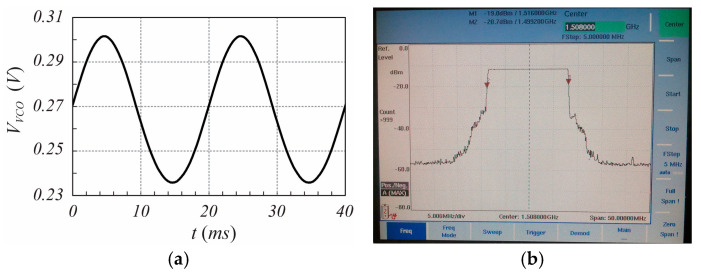
Time behavior of the VCO input signal enforced by the PLL board (**a**), spectrum of the modulated RF signal measured through a Willtek 9101 Handheld Spectrum Analyzer after a divide-by-16 prescaler module (**b**).

**Figure 10 sensors-25-02765-f010:**
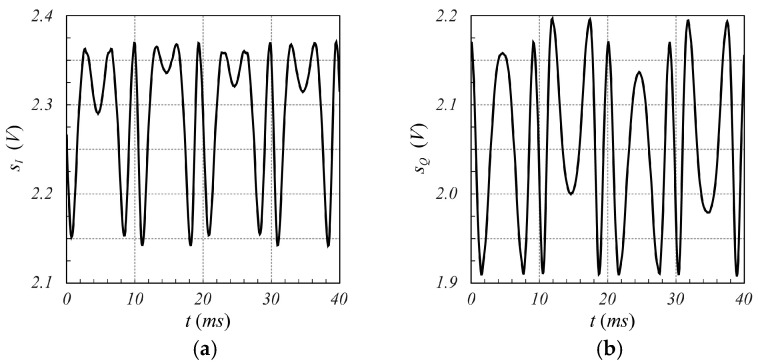
Time behavior of the signals (**a**) $s_{I}\left(t\right)$ and (**b**) $s_{Q}\left(t\right)$ provided by the SFMCW radar while the oscillating motion of a panel located 1.0 m far from the radar antennas is measured. A slightly different offset value between the two signals, due to the intrinsic asymmetry between the two I and Q channels of the radar, can be observed.

**Figure 11 sensors-25-02765-f011:**
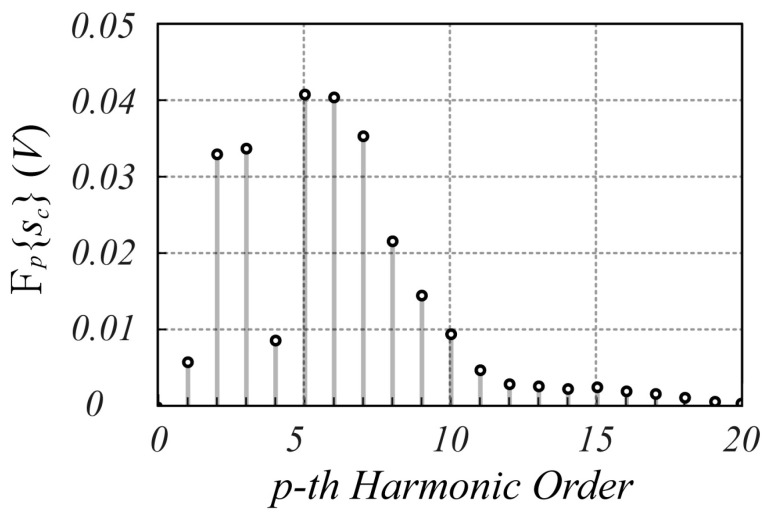
Spectrum of the $s_{c}\left(t\right)$ signal provided by the SFMCW radar during the oscillating motion of a panel located 1.0 m far from the radar antennas. The average value of the signals is subtracted to eliminate the DC offset.

**Figure 12 sensors-25-02765-f012:**
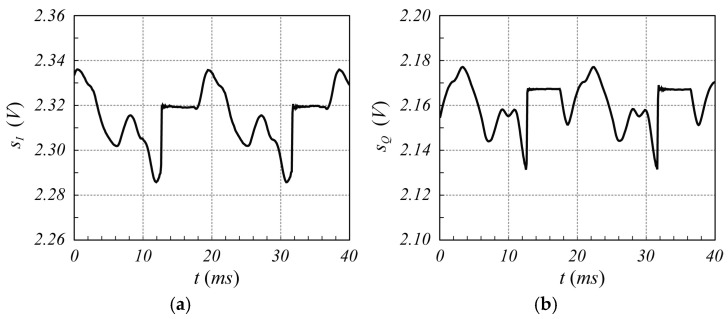
Time behavior of the signals (**a**) $s_{I}\left(t\right)$ and (**b**) $s_{Q}\left(t\right)$ provided by a chirp FMCW radar monitoring a metal sphere of 30 cm diameter with its center 1.0 m far from the radar antennas. Chirp characteristics: $T_{r}=14.4\;ms$, $T_{idle}=5.4\;ms$. Acquisition time window = 12.4 ms.

**Figure 13 sensors-25-02765-f013:**
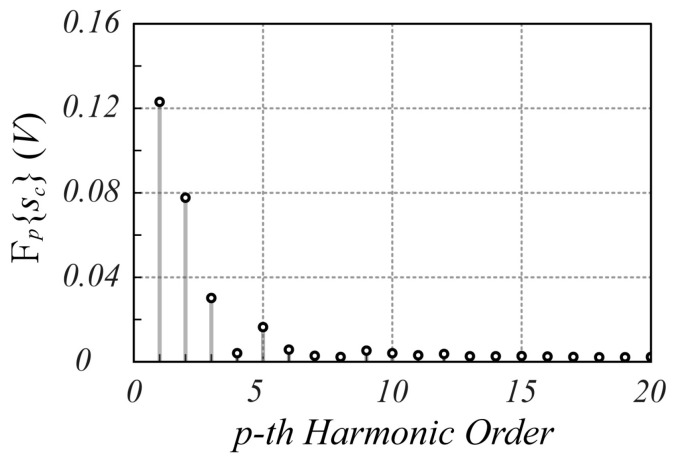
Spectrum of the $s_{c}\left(t\right)$ signal provided by a chirp FMCW radar monitoring a metal sphere of 30 cm diameter with its center 1.0 m far from the radar antennas. Chirp characteristics: $T_{r}=14.4\;ms$, $T_{idle}=5.4\;ms$. Acquisition time window = 12.4 ms. The average value of the signals is subtracted to eliminate the DC offset.

**Figure 14 sensors-25-02765-f014:**
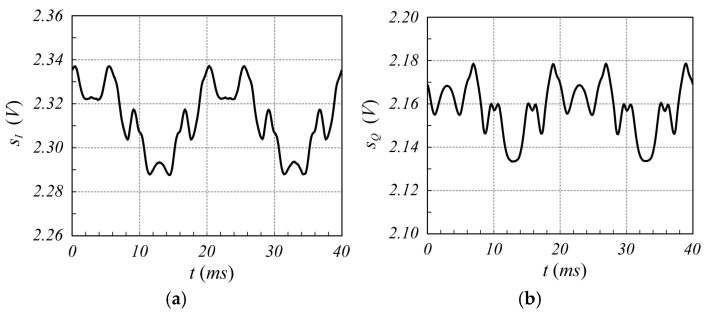
Time behavior of the signals (**a**) $s_{I}\left(t\right)$ and (**b**) $s_{Q}\left(t\right)$ provided by an SFMCW radar monitoring a metal sphere of 30 cm diameter with its center 1.0 m far from the radar antennas. Frequency of the VCO sinewave = 50 Hz.

**Figure 15 sensors-25-02765-f015:**
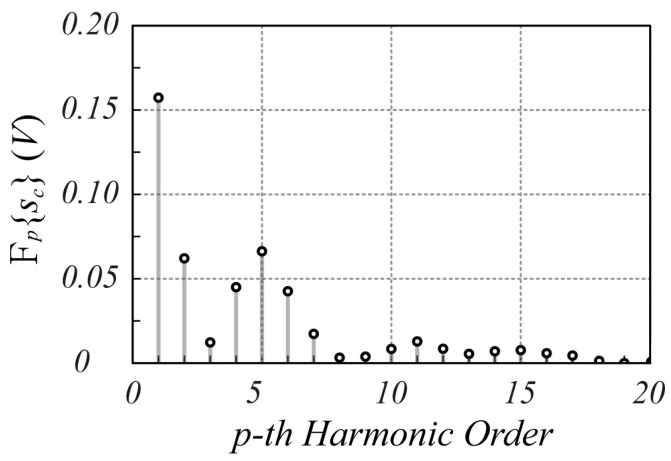
Spectrum of the $s_{c}\left(t\right)$ signal provided by the SFMCW radar monitoring a metal sphere of 30 cm diameter with its center 1.0 m far from the radar antennas. Frequency of the VCO sinewave = 50 Hz. The average value of the signals is subtracted to eliminate the DC offset.

**Figure 16 sensors-25-02765-f016:**
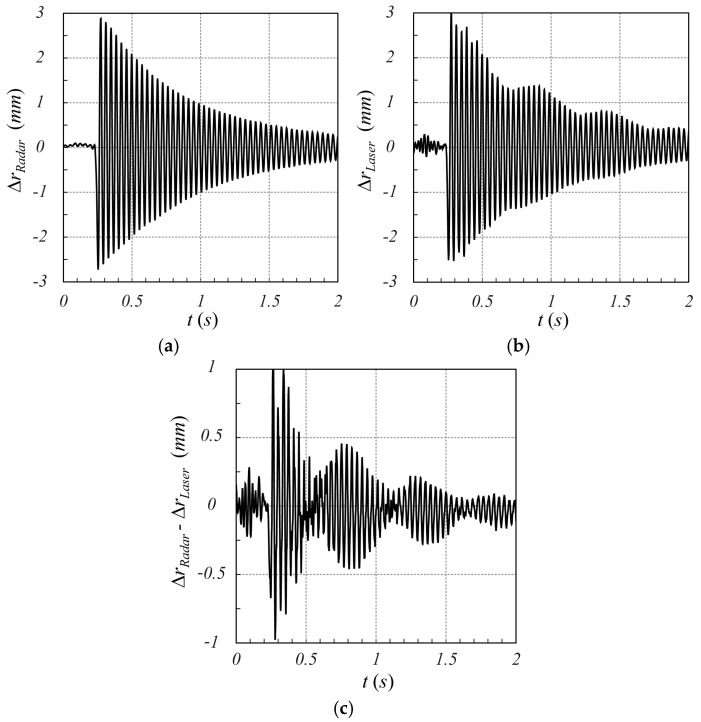
Time behavior of the panel movement measured by SFMCW (**a**) and laser sensor (**b**). In (**c**), the error between the two measurements is also reported. A good agreement between the two signals can be appreciated. The local displacement due to the higher-order mechanical mode detected by the laser sensor can be observed. The overall mean relative error between the two signals is about 0.1 mm.

## Data Availability

Data are available upon request.

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
