# Peer review of "A Novel FMCW Radar Scheme with Millimeter Motion Detection Capabilities Suitable for Cardio-Respiratory Monitoring"

_sensors, 2025, doi:10.3390/s25092765_

Round 1

Reviewer 1 Report

Comments and Suggestions for Authors

The core of this paper is to propose using a sine wave as the VCO drive signal, but how does it demonstrate low complexity? The title of the article is inappropriate.

The description in line 66 and the description of Figure 1(b) are not accurate. There can be an idle time window between chirps.

This paper does not conduct experiments on the accuracy of respiration and heart rate measurements, which mismatches the content and title of the article.

Additionally, there is no direct comparison between the proposed SFMCW system and the traditional FMCW system in terms of performance. The experimental section is inadequate and cannot fully substantiate the performance of the proposed new system.

If a new modulation method is proposed specifically for vital sign detection, it should clearly introduce the issues associated with traditional FMCW radar in vital sign detection. However, this part has not been clearly elaborated upon. 

Comments on the Quality of English Language

There are no major issues with the English.

Reviewer 2 Report

Comments and Suggestions for Authors

This is a good piece of work, and I am suggesting to publish it in your journal.

The strengths of the research work regarding:

  1. Theoretical Foundation

The research work presents a fundamentally different method to to FMCW radar modulation using sinusoidal frequency modulation (SFMCW) rather than using triangular or sawtooth patterns, which represents genuine innovation in the field.

  1. Mathematical Model

The authors developed a mathematical framework (equations 4-15) that extensively explains the theoretical basis of their method, including detailed signal analysis using Bessel functions to model the behavior of the system.

  1. Problem Statement:

The authors identified the limitations in conventional FMCW radar systems (synchronization requirements and signal distortion issues) and presented a solution to address these problems.

  1. Implementation

Practical hardware implantation with commercial components and custom signal processing with presenting the feasibility.

  1. Results validation

Both numerical simulation and hardware experiments are given in the research work. Moreover, electromagnetic exposure analysis for the sake of safter are presented.

A minor suggestion is, if possible, to make a comparison table of SFMCW against FMCW including hardware complexity, signal processing requirements, performance metrics, operational constraints, and implementation costs.

Note: In page 10, line 279; “the maximum EM exposure limits proposed in ICNIRP guidelines (10 𝑊/𝑚2 at 24 GHz) [20].” It is better to be “the maximum human body EM exposure limits proposed in ICNIRP guidelines (10 𝑊/𝑚2 at 24 GHz) [20].

Reviewer 3 Report

Comments and Suggestions for Authors

The manuscript is well written and detailed, and the current content is presented well. My primary concern is that additional supporting evidence is needed for some of the statements and conclusions that are made.  I have done vital signs detection in several different modalities, and reviewed many related papers.  In the past, I have not been concerned with synchronization procedures in the performance of the radar.  I have also flipped from sawtooth, triangle, and sinusoidal modulation and have not seen a noticeable difference in performance.  As such, while the details provided here are useful (details analysis of what to expect), you have not made a convincing, data-driven argument for sinusoidal modulation.  My thoughts for improvement and therefore adoption by others, are the following.  

  • Use the current damped plate to show the difference in performance between the sawtooth modulation (with and without synchronization) versus the sinusoidal modulation.  Quantify the problems caused by the lack of synchronization in sawtooth mode and how this improves with sinusoidal modulation.
  • Study the differences in the harmonic order (Fig. 3) between modulation schemes.  How is the spectral power distributed among the harmonics?
  • Related to the last comment, how is the range resolution impacted between modulation schemes?  
  • Show a difference plot between the data sets for Fig. 9.

There are always tradeoffs between different modalities, and while you state the benefit for this modulation, it was not quantified.  I also would like to know how other parameters are impacted, like range resolution and determining the range bin of the target. 

What is provided is well done, but dive deeper into the pros and cons by quantifying them. 

Round 2

Reviewer 1 Report

Comments and Suggestions for Authors

The manuscript currently lacks a comprehensive comparison with existing studies in this field. The authors need to demonstrate the differences and specific improvements offered by this study compared to prior research, highlighting its novelty explicitly.

Additionally, the authors should discuss whether the SFMCW approach can fully replace the traditional LFMCW radar technology. A detailed discussion of the limitations and constraints of the proposed SFMCW technique is also recommended to provide a balanced evaluation of its practical applicability and potential for future development.

Rodriguez, D.; et al. SMCW Radar for Low IF Sensing Applications. 2022 IEEE Topical Conf. on Wireless Sensors (WiSNet), 2022, pp. 22–25.​

Xu, Z.; et al. A Novel Demodulation Algorithm for Micro-Displacement Measurement Based on FMCW Sinusoidal Modulation. Photonics, 2023, 10(12): 1196.​

Reviewer 3 Report

Comments and Suggestions for Authors

The authors addressed the concerns of the first review.  The added specificity removes any ambiguity, and they supported their findings with a data-driven approach.  Very well done, and I would publish as is.  

One mild comment is that including a symbol legend in Figures 3 and 4 would help.  I found myself jumping between the two figures and captions.  However, the detailed figure captions are clear and appreciated.  
